# Association of Plasma Cortisol Levels with Gestational Age and Anthropometric Values at Birth in Preterm Infants

**DOI:** 10.3390/ijerph191811448

**Published:** 2022-09-11

**Authors:** Masako Aoki, Tatsuhiko Urakami, Nobuhiko Nagano, Ryoji Aoki, Ichiro Morioka

**Affiliations:** Department of Pediatrics and Child Health, Nihon University School of Medicine, Tokyo 173-8610, Japan

**Keywords:** anthropometric value, birth head circumference, birth weight, cortisol, preterm infant, gestational age, small-for-gestational age

## Abstract

There are no study reports to clarify the association between gestational age (GA) or anthropometric values at birth, and plasma cortisol levels in the blood of preterm infants at birth and at one month of age. This hospital-based retrospective cohort study included infants born at <37 weeks’ gestation between 2019 and 2021. First, the association between plasma cortisol level and GA or anthropometric values at birth (birth weight standard deviation score [SDS], birth length SDS, and birth head circumference SDS) was identified by regression and multiple regression analyses. Second, plasma cortisol levels in the umbilical cord at birth and at one month of age were compared between small-for-gestational age (SGA) and non-SGA infants. Sixty-one preterm infants were enrolled (SGA: 24 and non-SGA: 37). Plasma cortisol levels at birth were significantly associated with GA. Plasma cortisol levels at one month of age were associated with GA and birth head circumference SDS. Plasma cortisol levels at birth were significantly higher in SGA than non-SGA (*p* = 0.010). GA was an independent determinant of plasma cortisol levels at birth. SGA infants had a high plasma cortisol level at birth; resulting in speculation that a high plasma cortisol level at birth may predict abnormal neurological outcomes.

## 1. Introduction

Cortisol is an essential hormone for the maintenance of human health. During the fetal period, cortisol promotes organ maturation, whereas excess cortisol suppresses organ development. During the fetal period, maternal active cortisol is converted into inactive cortisone by 11β-hydroxysteroid dehydrogenase-2 (11βHSD2) in the placenta to prevent the influx of maternal active cortisol into the fetus. However, under stresses, such as poor nutrition and hypoxia, 11βHSD2 activity in the placenta is suppressed and maternal active cortisol flows into the fetus [1]. In preterm infants, 11βHSD2 activity is generally low, and maternal active cortisol flows directly into the infant via the placenta [2]. Elevated fetal cortisol is associated with cardiovascular, metabolic, and endocrine disorders in adulthood in some animal studies [3,4]. 

Fetal growth parameters, such as the body weight standard deviation score (SDS) and head circumference SDS, have been used as indicators to assess poor intrauterine nutrition [5]. Previous studies examined the relationship between cortisol levels and prematurity, small-for-gestational-age (SGA), and some neonatal diseases [6,7,8,9,10,11,12]. Iwata et al. reported that salivary cortisol levels in infants born at 31–39 weeks’ gestational age (GA) were higher in SGA infants than in non-SGA infants within 5 days of age and lower in SGA infants than in non-SGA infants within 14 days of age [6]. Grunau et al. evaluated plasma cortisol levels at 8–18 months of age in full-term or preterm infants born at <28 weeks’ gestation, and found higher plasma cortisol levels in preterm infants than in full-term infants [7]. Cortisol levels in preterm infants have also been associated with neonatal complications, such as chronic lung disease (CLD) [8,9], respiratory distress syndrome (RDS) [10,11], and late-onset circulatory collapse (LCC) [12]. Although there are a few reports regarding cortisol levels in neonates, no study has evaluated the association between plasma cortisol levels and GA or anthropometric parameters measured at birth in preterm infants. This study aimed to clarify the association between GA or anthropometric values at birth and plasma cortisol levels in umbilical cord blood at birth and at one month of age in preterm infants (Study 1) and to compare plasma cortisol levels between small-for-gestational age (SGA) and non-SGA infants (Study 2). 

## 2. Materials and Methods

### 2.1. Eligible Infants 

This retrospective cohort study was approved by the Ethics Committee of Nihon University Itabashi Hospital (approval no: RK-190910-3; date: 20 November 2019). All preterm infants born by cesarean section at Nihon University Itabashi Hospital between September 2019 and June 2021 were enrolled in the study. Written informed consent was provided by the parents. Blood samples of the infants at birth (umbilical cord blood) and at 1 month of age were obtained for routine laboratory tests for various medical reasons. After the relevant laboratory tests were performed, the residual blood samples were used to measure plasma cortisol levels. The exclusion criteria were prenatal diagnosis of monochorionic twins, severe congenital heart malformation, and chromosomal abnormalities. 

Preterm infants, who were born between 22 and 36 weeks after conception, were then divided into three groups based on Japanese neonatal anthropometric charts for GA at birth [13]: birth weight (BW) less than the 10th percentile (small-for-gestational age [SGA]), BW between the 10th and 90th percentile (appropriate-for-gestational age [AGA]), and BW more than the 90th percentile (large-for-gestational age [LGA]).

### 2.2. Blood Sample Collection and Plasma Cortisol Measurement

Blood samples were collected from the umbilical vein immediately after birth and from vein puncture at 1 month of age (median [minimum-maximum]: 28 (25–49) days after birth), and blood samples at 1 month of age were collected between 8 and 9 a.m. before feeding. Plasma cortisol levels were measured using the electrochemiluminescence immunoassay method, as previously reported [14]. The difference between the plasma cortisol levels at 1 month of age and birth (umbilical cord blood) was defined as Δ cortisol.

### 2.3. Nutrition Policy from Birth to One-Month of Age

Shortly after birth, a 10% sugar solution was initiated at a water quantity (WQ) of 50–60 mL/kg/day. Thereafter, the WQ was increased by approximately 10 mL/kg/day, considering daily weight gain or loss, urine output, and insensible excretion. In the case of infusion alone, the WQ was maintained at 100–120 mL/kg/day. Depending on the infant’s condition, enteral feeding (exclusive breast feeding) was initiated, the infusion was tapered and the feeding volume was maintained at a WQ of 130 mL/kg/day with enteral feeding alone. After 35 weeks of corrected GA, infants were fed independently if possible.

### 2.4. Clinical Characteristics and Disease Definitions 

Maternal clinical factors included maternal age at birth, pre-pregnancy body mass index (BMI), weight gain during pregnancy, maternal height, placental weight, multiple births, the development of hypertensive disorder of pregnancy (HDP) and gestational diabetes mellitus (GDM), presence of chorioamnionitis, and prepartum intramuscular betamethasone use (two intramuscular injections of 12 mg are administered 24 h apart to pregnant women at high risk of preterm birth before 33 weeks and 6 days according to our management policy). HDP is defined as hypertension (blood pressure ≥ 140/90 mmHg) during pregnancy [15]. GDM is defined as glucose intolerance with onset or first recognition during pregnancy. GDM was diagnosed when an oral glucose tolerance test (OGTT) was performed and one or more of the following criteria were met: fasting blood glucose ≥ 92 mg/dL, 1-h value ≥ 180 mg/dL, or 2-h value ≥ 153 mg/dL in the 75-g OGTT [16,17] Regarding multiple births, monochorionic twins were excluded because the umbilical cord blood was affected due to blood flow traffic between the twins. Chorioamnionitis was defined as placental pathology with a grade 1–3 Blanc classification [18]. Neonatal clinical factors included sex, GA, BW, birth length (BL), birth head circumference, BW SDS, BL SDS, birth head circumference SDS, and development of RDS, CLD, and LCC. RDS was defined as the presence of reticular granular shadows on plain chest radiographs [19]. CLD was defined as respiratory distress symptoms requiring oxygen supply continuing beyond 28 days of age due to lung abnormalities, excluding congenital respiratory malformations, based on the Japanese criteria [20]. LCC was defined as a sudden disorder that manifested as hypotension and oliguria at one week or more after birth [12]. Patent ductus arteriosus was defined when surgical ligation was performed. Sepsis was defined as an elevated serum C-reactive protein level, bacterial pathogen detected in culture, and antibiotic treatment. Premature retinopathy was defined when photocoagulation was required. Intracranial hemorrhage and periventricular leukomalacia were diagnosed using cranial ultrasonography and magnetic resonance imaging. Necrotizing enteritis was defined based on the pathological findings after surgery. Meconium disease was defined when treated by contrast agent [21].

### 2.5. Statistical Analysis 

Study 1: To clarify the association between GA or anthropometric values (BW SDS, BL SDS, and head circumference SDS) at birth with plasma cortisol levels at birth and at 1 month of age in preterm infants, regression analysis was performed to linearly compare GA or each anthropometric SDS at birth and plasma cortisol levels at birth and at 1 month of age. Δ cortisol levels, coefficient of determination (R^2^), and *p*-values were calculated. Multiple regression analysis was performed using plasma cortisol levels at birth and at 1 month of age and Δ cortisol levels as objective variables, and GA and each anthropometric SDS at birth as explanatory variables. 

Study 2: To compare the following between SGA and non-SGA infants, the Mann—Whitney U test and Fisher’s exact test were performed: maternal and neonatal clinical factors, cord blood plasma cortisol levels, plasma cortisol levels at 1 month of age, and Δ cortisol levels. 

Statistical analyses were performed using JMP ver14.0 (JMP Statistical Discovery LLC, Tokyo, Japan), and *p* < 0.05 was considered significant.

## 3. Results

### 3.1. Study 1

#### 3.1.1. Clinical Characteristics

During the study period, 61 (SGA: 24, non-SGA: 37) preterm infants were enrolled. Of the non-SGA infants, 35 were AGA and 2 were LGA. The clinical characteristics of the enrolled infants and their mothers are presented in Table 1. Since six mothers with four di- and two tri-chorionic births were included, the clinical characteristics of 53 mothers are shown.

#### 3.1.2. Plasma Cortisol Levels at Birth and at 1 Month of Age

Figure 1 shows the relationship between GA and plasma cortisol levels at birth and at 1 month of age. GA was significantly correlated with plasma cortisol levels at birth and at 1 month of age (R^2^ = 0.154, *p* = 0.0018 and R^2^ = 0.146, *p* = 0.0024, respectively).

#### 3.1.3. Association between Plasma Cortisol Levels and GA, and Anthropometric Values at Birth

Plasma cortisol levels in umbilical blood and at 1 month of age, and Δ cortisol levels were significantly positively correlated with GA (Table 2A–C). Δ cortisol levels were significantly correlated with head circumference SDS (R^2^ = 0.078, *p* = 0.029; Table 2C). In multivariate analyses, plasma cortisol levels of umbilical blood and Δ cortisol were significantly positively correlated with GA (*p* < 0.05, Table 3A–C). Plasma cortisol levels at 1 month of age exhibited a significantly positive correlation with not only GA, but also birth head circumference SDS (partial correlation coefficient = 2.533, *p* = 0.015).

### 3.2. Study 2

#### 3.2.1. Clinical Characteristics 

Mothers who delivered at least one SGA infant were included in the SGA group. Maternal complications, such as multiple births, HDP, GDM, and chorioamnionitis were not significantly different between the two groups (Table 4). No significant difference was found between the two groups regarding maternal betamethasone use (SGA 69.6% vs. non-SGA 70.0%, *p* = 1.000). No significant difference between the two groups was found with respect to the timing of betamethasone use (*p* = 0.113). In the SGA group, fetal distress was the most common cause for delivery (13/23 cases, 56.5%). In the non-SGA group, threatened preterm delivery was the most common cause for delivery (21/30 cases, 70.0%).

As expected, BW, BL, and birth head circumference were significantly lower in the SGA group than in the non-SGA group. No significant differences were found in the GA (Table 4).

#### 3.2.2. Plasma Cortisol Levels

Plasma cortisol levels at birth and at 1 month of age were plotted based on GA in SGA and non-SGA infants (Figure 1). A comparison of plasma cortisol levels between SGA and non-SGA infants is presented in Table 5. Plasma cortisol levels at birth were significantly higher in the SGA group than in the non-SGA group (*p* = 0.010). However, there were no significant differences in plasma cortisol levels at 1 month of age and Δ cortisol levels between the two groups. 

## 4. Discussion

In the present study, we observed that in preterm infants, plasma cortisol levels at birth and at 1 month of age were positively correlated with GA. Importantly, plasma cortisol levels at 1 month of age were positively correlated with head circumference SDS at birth. SGA infants had significantly higher plasma cortisol levels at birth than non-SGA infants; however, this significant difference disappeared at 1 month of age.

### 4.1. Maternal Steroid Administration and Fetal Cortisol Levels

Most of the cortisol in maternal blood is inactivated by 11βHSD2 in the placenta to maintain fetal blood cortisol levels at approximately 10% of the maternal levels [22]. Hence, the fetus develops in an environment with low cortisol concentrations before delivery, but the high concentrations affect the hypothalamic–pituitary–adrenal axis (HPA-axis) [23]. Thus, the regulation of fetal cortisol concentration involves placental 11βHSD2 [3] and the fetal HPA axis [23]. Prepartum synthetic steroid administration, intrauterine infection, and HDP are also known to affect fetal cortisol concentrations [22]. Synthetic steroids (betamethasone or dexamethasone) are commonly used for pulmonary maturation in preterm infants before delivery, and are not inactivated in the placenta. However, they are known to accumulate in the fetus and affect fetal cortisol levels [22]. In Study 1, because approximately 70% of the mothers received betamethasone, plasma cortisol levels were affected in some infants. In Study 2, the use and timing of maternal betamethasone did not differ significantly between the SGA and non-SGA groups.

### 4.2. Factors Affecting Plasma Cortisol Levels and Physique

GA is known to affect fetal cortisol levels [24,25]. Similar results were obtained in Study 1. Importantly, we found that plasma cortisol levels at 1 month of age were associated with birth head circumference SDS, as well as GA. There have been several reports regarding physique and cortisol levels [7,24,26,27], but they vary in background; cortisol specimens included hair, serum, and saliva. The GA and age at the time of measurement were not considered.

In fetal growth restriction, blood flow to the limbs decreases, height decreases, and weight gain decreases. However, vital organs, such as the brain and adrenal glands, are prioritized [28]. SGA infants with severely small head circumferences do not maintain their height and weight [29,30]. As such, height and weight are affected earlier than the head circumference. When the head circumference can no longer be maintained, the adrenal function, as well as the brain also declines. Cortisol levels might be more closely associated with head circumference, rather than weight or height. There is a possibility that studies conducted with larger cohorts may reveal an association between BW or BL and plasma cortisol levels. However, because it does not affect the adrenal glands at the time at which BL and BW are affected, the association between BL and BW is not clinically significant. Plasma cortisol levels in the first month of life reflect an infant’s adrenal function. Infants with small head circumferences may have low plasma cortisol levels at 1 month of age. Further studies are required to confirm this hypothesis.

### 4.3. Plasma Cortisol Levels in Preterm SGA Infants

SGA infants had significantly higher plasma cortisol levels at birth than non-SGA infants; fetal distress was the most common cause of preterm birth in SGA infants, which was more common than in non-SGA infants. Previous studies reported that the HPA axis is affected by fetal distress, undernutrition, maternal infection, and strict diet [22,31]. Excessive fetal cortisol exposure may also induce fetal growth restriction and preterm birth [32]. In a previous report, salivary cortisol levels within 5 days of age in SGA at 31–39 GA were significantly higher than those in non-SGA [6] In the present study, SGA preterm infants were considered to be exposed to low nutrition or fetal distress, which caused HPA-axis activity and high plasma cortisol levels at birth. Furthermore, this result was implicated to be important in relation to long-term neurological outcome. In the Cochrane library, Doyle et al. reported long-term neurological outcome in preterm infants who had steroids at <8 days of age to prevent or treat CLD in meta-analyses; resulting in the view that dexamethasone treatment at an early age was associated with increased risk of an abnormal neurological examination, developmental delay, and cerebral palsy [33]. A high plasma cortisol level at birth in preterm SGA infants may, therefore, be associated with abnormal neurological outcome. Plasma cortisol level at birth may be a predictive factor for long-term abnormal neurological outcome. Further prospective cohort studies are required to investigate long-term neurological outcomes in preterm SGA infants with a high plasma cortisol level.

Our results revealed that plasma cortisol levels at 1 month of age were not significantly different between the SGA and non-SGA groups. Iwata et al. reported that cortisol levels in saliva were higher in the non-SGA group than in the SGA group after 14 days of age [6]. The reason for this might be that their report involved a mixture of preterm and full-term infants. Furthermore, there was no consideration of GA in the comparison between SGA and non-SGA infants. SGA infants have high plasma cortisol levels at birth and fetal distress. The HPA axis, and placental 11βHSD2, associated with low nutrition and fetal distress, are profoundly involved, but this effect disappears at 1 month of age. 

### 4.4. Limitations

Our study had some limitations. First, because this was a single-center study with a short duration, the number of infants in the cohort was limited. Second, this study included all preterm infants born by cesarean section with a high percentage of prenatal dexamethasone administrations to pregnant women, which was a major limitation. Słabuszewska-Jóżwiak et al. reported that there was no difference in umbilical cord blood cortisol levels between cesarean and vaginal delivery, but umbilical cord blood cortisol levels were significantly lower in scheduled cesarean section delivery than in vaginal delivery, possibly due to less distress involved in uterine contractions during delivery compared to vaginal delivery [34]. In this study, the differences in delivery were not examined. The activation of the HPA-axis is preparation for birth and the optimal and physiological reactions occur in a vaginal birth. Further studies using preterm infants born by vaginal delivery are needed to confirm our results.

## 5. Conclusions

GA is an independent determinant of plasma cortisol levels at birth. SGA infants had a high plasma cortisol level at birth; resulting in speculation that a high plasma cortisol level at birth may be associated with abnormal neurological outcome. Further follow-up studies using a large cohort are required to confirm this result.

## Figures and Tables

**Figure 1 ijerph-19-11448-f001:**
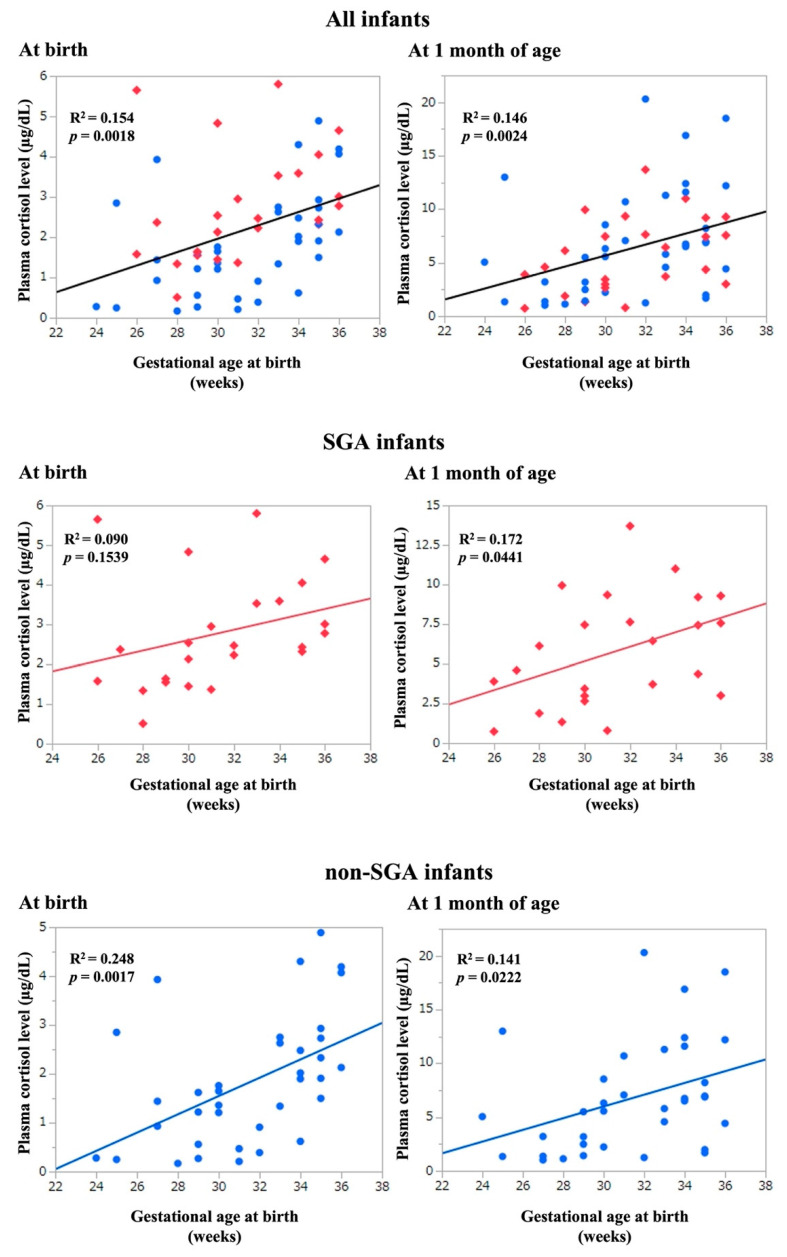
Gestational age at birth and plasma cortisol levels at birth or at 1 month of age. Red rhombus: small-for-gestational age (SGA), Blue circle: non-SGA.

**Table 1 ijerph-19-11448-t001:** Clinical characteristics of mothers and infants.

**Mothers**	***N* = 53**
Age at delivery, years	31 (24–36)
Pre-pregnancy BMI, kg/m^2^	21.5 (15.6–33.8)
Weight gain during pregnancy, kg	5.9 (0.0–24.4)
Height, cm	156.9 (144.6–172.0)
Placenta weight, g	348 (136–804)
Di- and tri-chorionic births, *n* (%)	6 (11.3)
Hypertensive disorders of pregnancy, *n* (%)	16 (30.2)
Gestational diabetes mellitus, *n* (%)	1 (1.9)
Chorioamnionitis, *n* (%)	28 (52.8)
Prepartum use of betamethasone, *n* (%)	37 (69.8)
Last use of betamethasone, days before delivery, *n* = 37	6 (0–35)
**Infants**	***N* = 61**
Male sex, *n* (%)	22 (36)
GA, weeks	31 (24–36)
BW, g	1424 (464–2834)
BW SDS	−0.91(−4.23–1.69)
BL, cm	39.0 (28.0–47.0)
BL SDS	−0.9 (−4.23–1.69)
Birth head circumstance, cm	28.8 (20.0–34.5)
Birth head circumstance SDS	−0.16 (−2.48–1.92)
Respiratory distress syndrome, *n* (%)	26 (42.6)
Chronic lung disease, *n* (%)	14 (23.0)
Late-onset circulatory collapse, *n* (%)	5 (8.2)
Patent ductus arteriosus, *n* (%)	0 (0)
Sepsis, *n* (%)	1 (1.6)
Retinopathy of prematurity, *n* (%)	2 (3.3)
Necrotizing enterocolitis, *n* (%)	0 (0)
Meconium disease, *n* (%)	2 (3.3)
Intraventricular hemorrhage, *n* (%)	0 (0)
Periventricular leukomalacia, *n* (%)	2 (3.3)

Data are shown as the median (range) or number (percentage); BMI, body mass index; BL, birth length; BW, birth weight; GA, gestational age at birth; SDS, standard deviation score.

**Table 2 ijerph-19-11448-t002:** Regression analysis.

**A. Cortisol Level at Birth**		
	**Coefficient of Determination**	***p*-Value**
GA, weeks	0.154	0.002
BW SDS	0.057	0.063
BL SDS	0.063	0.051
Birth head circumference SDS	0.015	0.344
**B. Cortisol Level at 1 Month of Age**		
	**Coefficient of Determination**	***p*-Value**
GA, weeks	0.146	0.002
BW SDS	0.014	0.370
BL SDS	0.010	0.453
Birth head circumference SDS	0.058	0.062
**C. Δ Cortisol**		
	**Coefficient of Determination**	***p*-Value**
GA, weeks	0.067	0.044
BW SDS	0.037	0.137
BL SDS	0.031	0.172
Birth head circumference SDS	0.078	0.029

BL, birth length; BW, birth weight; CI, confidence interval; GA, gestational age at birth; SDS, standard deviation score.

**Table 3 ijerph-19-11448-t003:** Multivariate regression analysis.

**A. Cortisol Level at Birth**		
	**Partial Correlation Coefficient (95% CI)**	***p*-Value**
GA, weeks	0.191 (0.092–0.290)	<0.001
BW SDS	−0.459 (−0.934–0.015)	0.058
BL SDS	−0.127 (−0.581–0.328)	0.580
Birth head circumference SDS	0.495 (−0.125–1.114)	0.115
**B. Cortisol Level at 1 Month of Age**		
	**Partial Correlation Coefficient (95% CI)**	***p*-Value**
GA, weeks	0.558 (0.236–0.881)	0.001
BW SDS	−1.151 (−2.700–0.398)	0.142
BL SDS	0.095 (−1.389–1.578)	0.899
Birth head circumference SDS	2.533 (0.514–4.552)	0.015
**C. Δ Cortisol**		
	**Partial Correlation Coefficient (95% CI)**	***p*-Value**
GA, weeks	0.367 (0.028–0.707)	0.034
BW SDS	−0.692 (−2.323–0.939)	0.399
BL SDS	0.221 (−1.341–1.783)	0.778
Birth head circumference SDS	2.039 (−0.876–4.165)	0.060

BL, birth length; BW, birth weight; CI, confidence interval; GA, gestational age at birth; SDS, standard deviation score.

**Table 4 ijerph-19-11448-t004:** Clinical characteristics of SGA and non-SGA groups.

**Mothers** ***N* = 53**	**SGA** ***N* = 23**	**Non-SGA** ***N* = 30**	***p*-Value**
Age at delivery, years	31 (26–36)	32 (24–36)	0.212
Pre-pregnancy BMI, kg/m^2^	22.2 (16.7–33.8)	20.9 (15.6–26.6)	0.049
Weight gain during pregnancy, kg	7.1 (1.0–24.4)	5.8 (0.0–14.8)	0.543
Height, cm	157.6 (149.0–172.0)	156.2 (144.6–168.0)	0.287
Placenta weight, g	311 (136–480)	420 (242–804)	<0.001
Di- and tri-chorionic births, *n* (%)	3 (13.0)	3 (10.0)	1.000
Hypertensive disorders of pregnancy, *n* (%)	12 (52.2)	4 (13.3)	0.006
Gestational diabetes mellitus, *n* (%)	1 (4.3)	0 (0.0)	0.434
Chorioamnionitis, *n* (%)	10 (43.5)	18 (60.0)	0.276
Prepartum use of betamethasone, *n* (%)	16 (69.6)	21 (70.0)	1.000
Last use of betamethasone, days before delivery, *n* = 37	14 (0–35), *n* = 16	4 (0–28), *n* = 21	0.113
**Infants** ***N* = 61**	**SGA** ***N* = 24**	**Non-SGA** ***N* = 37**	
Male sex, *n* (%)	11 (45.8)	11 (29.7)	0.068
GA, weeks	31 (26–36)	32 (24–36)	0.796
BW, g	1051 (464–1900)	1575 (578–2834)	<0.001
BW SDS	−2.5 (–4.23–−1.29)	−0.23 (−1.27–1.69)	<0.001
BL, cm	36.4 (28.0–46.5)	40.5 (30.0–47.0)	0.001
BL SDS	−2.2 (−4.17–0.33)	−0.31 (−1.89–1.71)	<0.001
Birth head circumstance, cm	26.7 (20.7–31.0)	29.2 (20.0–34.5)	0.013
Birth head circumstance SDS	−1.0 (−2.48–0.50)	0.14 (−1.07–1.92)	<0.001
Respiratory distress syndrome, *n* (%)	11 (45.8)	15 (40.5)	0.793
Chronic lung disease, *n* (%)	7 (29.2)	7 (18.9)	0.370
Late-onset circulatory collapse, *n* (%)	3 (12.5)	2 (5.4)	0.373
Sepsis, *n* (%)	0 (0)	1 (2.7)	1.000
Retinopathy of prematurity, *n* (%)	1 (4.2)	1 (2.7)	1.000
Meconium disease, *n* (%)	1 (4.2)	1 (2.7)	1.000
Periventricular leukomalacia, *n* (%)	1 (4.2)	1 (2.7)	1.000

Data are shown as the median (range) or number (percentage). BMI, body mass index; BL, birth length; BW, birth weight; GA, gestational age at birth; SDS, standard deviation score.

**Table 5 ijerph-19-11448-t005:** Comparison of plasma cortisol levels between SGA and non-SGA infants.

	All Infants *N* = 61	SGA *N* = 24	Non-SGA *N* = 37	*p*-Value
At birth, μg/dL	2.02 (0.17–5.80)	2.45 (0.51–5.80)	1.62 (0.17–4.89)	0.010
1 month of age, μg/dL	5.80 (0.74–20.3)	5.37 (0.74–13.7)	5.80 (1.03–20.3)	0.816
Δ cortisol, μg/dL	4.21 (−4.91–19.39)	2.28 (−4.91–11.47)	4.61 (−2.55–19.39)	0.181

Data are shown as medians (ranges).

## Data Availability

The data supporting the findings of this study are available from the corresponding author upon reasonable request.

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
