# Peer review of "Association of Plasma Cortisol Levels with Gestational Age and Anthropometric Values at Birth in Preterm Infants"

_ijerph, 2022, doi:10.3390/ijerph191811448_

Round 1
Reviewer 1 Report
The assessed manuscript has limitations that were stated by the authors themselves in the Discussion: it is a single center study with a limited number of cases, cesarean deliveries, with a high percentage of prenatal dexamethasone administration to pregnant mothers.
The conclusions - despite the significance of the differences in the compared sets after birth - are a speculative hypothesis. It is proven that the activation of the hypothalamic-pituitary axis before birth - including cortisol - is preparation for birth. There is a maturation of the lungs, surfactant factor, antioxidant systems, etc. The optimal and physiological reaction is a vaginal birth - when cortisol levels are higher than in a cesarean delivery. As the authors note – further studies are needed to confirm their hypothesis.Author Response
Response to Reviewer #1:
The assessed manuscript has limitations that were stated by the authors themselves in the Discussion: it is a single center study with a limited number of cases, cesarean deliveries, with a high percentage of prenatal dexamethasone administration to pregnant mothers. The conclusions - despite the significance of the differences in the compared sets after birth - are a speculative hypothesis. It is proven that the activation of the hypothalamic-pituitary axis before birth - including cortisol - is preparation for birth. There is a maturation of the lungs, surfactant factor, antioxidant systems, etc. The optimal and physiological reaction is a vaginal birth - when cortisol levels are higher than in a cesarean delivery. As the authors note – further studies are needed to confirm their hypothesis.
Response: We agreed with your comments that the optimal and physiological reaction is a vaginal birth. Therefore, we strongly stated this fact as a major limitation as follows.
Discussion
Lines 296–298: Second, in a major limitation, this study included all preterm infants born by cesarean section with a high percentage of prenatal dexamethasone administrations to pregnant women.
Lines 303–306: The activation of HPA-axis is preparation for birth and the optimal and physiological reactions occur in a vaginal birth. Further studies using the preterm infants born by vaginal delivery are needed to confirm our results.

Reviewer 2 Report
Dear authors,
thank you for the manuscript on the association of cortisol levels with GA and anthropometric values. Although the design of the study and the results are presented nicely, I have some major comments:
1. In the introduction (line 34-36) you state that elevated fetal cortisol is associated with cardiovascular, metabolic, and endocrine disorders in adulthood. However, the reader would primarily think that this refers to humans, only on reading the cited articles (3,4) it gets clear that these date come from sheep. This is nowhere mentioned in your manuscript, nor is the possible transition to humans, and limitations discussed. Also it would be necessary to discuss the influance of additional risk factors during infancy, adollescence, and early adulthood, that could affect the developement of these diseases. And last but not least, the nearly impossibility to conduct a study to proof your suggestion, would be necessary.
2. There are several wrong or hardly understandable phrasings/citations:
- page 8, lines 221-222: ... regarding physique and cortisol levels
- page 8, line 239: ..., fetal dysfunction
- page 8, lines 242-243: ... salivary cortisol levels within 5 days of conception
These should be rewriten, and it needs to be more clearly described whether cortisol levels in the preterms, or the mothers was meant.
3. Figure 1 (A,B) should be devided into 2 figures. And R2- and p-values should be calculated seperatly for the 2 groups SGA and non-SGA.
4. Discussion: The discussion should be completly rewriten. It should be better structured in view of the (new) results that the authors and then compared in relation to prvious studies.
5. Conclusion: The conlusion is not a conclusion but just a short summary of the 3 major findings of the authors study.
The conclusion should summarize, what the reader could learn from the present study.
However, most important is the question: Can this study add anything to improve the care for preterm infants, and how could this improvement also improve the outcome.
If not this, than at least a proposal, what kind of studies should be initiated in the future to answer these 2 questions.
If these last issues cannot be solved, than it seems that there is not really much novel information that would justify publication.
Author Response
Response to Reviewer #2:
Reviewer 2
Thank you for the manuscript on the association of cortisol levels with GA and anthropometric values. Although the design of the study and the results are presented nicely, I have some major comments.
Response: Thank you for your review for us.
- In the introduction (line 34-36) you state that elevated fetal cortisol is associated with cardiovascular, metabolic, and endocrine disorders in adulthood. However, the reader would primarily think that this refers to humans, only on reading the cited articles (3,4) it gets clear that these date come from sheep. This is nowhere mentioned in your manuscript, nor is the possible transition to humans, and limitations discussed. Also it would be necessary to discuss the influance of additional risk factors during infancy, adollescence, and early adulthood, that could affect the developement of these diseases. And last but not least, the nearly impossibility to conduct a study to proof your suggestion, would be necessary.
Response: We thank you for your good suggestions.
You are correct. The data in references #3 and #4 were from the animal studies. We added “in some animal studies” (Line 38). We also deleted “Importantly” in this sentence.
We deepely agreed with your comments that it is impossible to show the association between elevated fetal cortisol and cardiovascular, metabolic, and endocrine disorders in adulthood in human. Therefore, we deleted “Second, cortisol in the fetus is associated with cardiovascular, metabolic, and endocrine disorders in adulthood; however, as this study was conducted from birth until the first month after birth, information regarding the long-term prognosis of infants was not available” in Discussion section.
- There are several wrong or hardly understandable phrasings/citations:
- page 8, lines 221-222: ... regarding physique and cortisol levels
Response: The citations for this sentence were changed from [25,26] to [7,24,26,27].
Lines 239–240: There have been several reports regarding physique and cortisol levels [7,24,26,27], but…….
- page 8, line 239: ..., fetal dysfunction
Response: “fetal dysfunction” was changed to “fetal distress” in text. Thank you.
- page 8, lines 242-243: ... salivary cortisol levels within 5 days of conception
These should be rewriten, and it needs to be more clearly described whether cortisol levels in the preterms, or the mothers was meant.
Response: We are sorry for our mistake. “5 days of conception” was changed to “5 days of age”.
- Figure 1 (A,B) should be devided into 2 figures. And R2- and p-values should be calculated seperatly for the 2 groups SGA and non-SGA.
Response: According to the reviewer suggestions, the figures were separated.
- Discussion: The discussion should be completly rewriten. It should be better structured in view of the (new) results that the authors and then compared in relation to prvious studies.
Response: We thank you for your suggestions. Discussions were structured with sub-heading title. We also added the discussion in clinical perspectives based on a previous report. Please see the answer for 5 below.
- Conclusion: The conlusion is not a conclusion but just a short summary of the 3 major findings of the authors study.
The conclusion should summarize, what the reader could learn from the present study.
However, most important is the question: Can this study add anything to improve the care for preterm infants, and how could this improvement also improve the outcome.
If not this, than at least a proposal, what kind of studies should be initiated in the future to answer these 2 questions. If these last issues cannot be solved, than it seems that there is not really much novel information that would justify publication.
Response: We added the some sentences with a citation in clinical perspectives as follows.
Lines 265–285: Also, this result implicated to be important in relation to the long-term neurological outcome. In cochrane library, Doyle et al. have reported long-term neurological outcome in preterm infants used steroids at < 8 days of age to prevent or treat CLD in meta-analyses; resulting that the dexamethasone treatment in early age was associated with increased risk of an abnormal neurological examination, developmental delay, and cerebral palsy [33]. A high plasma cortisol level at birth in preterm SGA infants may be associated with abnormal neurological outcome, therefore, A plasma cortisol level at birth may be a predictive factor for long-term abnormal neurological outcome. Further prospective cohort studies are required to investigate a long-term neurological outcome in preterm SGA infants with a high plasma cortisol level.
[33] Doyle, L.W.; Ehrenkranz, R.A.; Halliday, H.L. Early (< 8 days) postnatal corticosteroids for preventing chronic lung disease in preterm infants. Cochrane Database Syst Rev. 2014: 5, CD001146.
Our conclusions were changes as follows.
Lines 308–311: GA is an independent determinant of plasma cortisol levels at birth. SGA infants had a high plasma cortisol level at birth; speculating that a high plasma cortisol level at birth may be associated with abnormal neurological outcome. Further follow-up studies using a large cohort are required to confirm this result.
We also changed the conclusion in abstract.
Lines 22–24: GA was an independent determinant of plasma cortisol levels at birth. SGA infants had a high plasma cortisol level at birth; speculating that a high plasma cortisol level at birth may predict abnormal neurological outcome.

Reviewer 3 Report
The authors preset a highly descriptive retrospective clinical cohort study on cortisol levels in preterm infants and its impact on anthropometric measures. The study does provide some novel findings on the change of cortisol levels from birth to one month of age between small for gestational age and average for gestational age infants.
Minor Concerns:
- The abstract does not clearly state the problem/need for the research, nor does it include a sufficient conclusion statement that interprets the findings
- Under Statistical analysis Study 2, the sentence should be re-written for clarity and possibly expanded into more than one sentence.
- In results, line 157, I believe there is a typo and you did not intend to write "i.e.".
- I find Table 2 and Table 3 difficult to read. There should be clearer separation of sections A., B., and C. Also, the heading for each should include "cortisol", so the reader can easily know what comparisons are being made.
- In the discussion, line 209 you wrote "post-growth". This should be corrected for clarity.
Author Response
Response to Reviewer #3:
Reviewer 3
The authors preset a highly descriptive retrospective clinical cohort study on cortisol levels in preterm infants and its impact on anthropometric measures. The study does provide some novel findings on the change of cortisol levels from birth to one month of age between small for gestational age and average for gestational age infants.
Response: Thank you for your review and your warm comments.
Minor Concerns:
- The abstract does not clearly state the problem/need for the research, nor does it include a sufficient conclusion statement that interprets the findings
Response: We changed the sentences in abstract as follows.
Lines 11–24: There are no study reports to clarify the association between gestational age (GA) or anthropometric values at birth, and plasma cortisol levels in the blood of preterm infants at birth and at 1 month of age…………………………………………………GA was an independent determinant of plasma cortisol levels at birth. SGA infants had a high plasma cortisol level at birth; speculating that a high plasma cortisol level at birth may predict abnormal neurological outcome.
We added the some sentences with a citation in clinical perspectives as follows.
Lines 265–285: Also, this result implicated to be important in relation to the long-term neurological outcome. In cochrane library, Doyle et al. have reported long-term neurological outcome in preterm infants used steroids at < 8 days of age to prevent or treat CLD in meta-analyses; resulting that the dexamethasone treatment in early age was associated with increased risk of an abnormal neurological examination, developmental delay, and cerebral palsy [33]. A high plasma cortisol level at birth in preterm SGA infants may be associated with abnormal neurological outcome, therefore, A plasma cortisol level at birth may be a predictive factor for long-term abnormal neurological outcome. Further prospective cohort studies are required to investigate a long-term neurological outcome in preterm SGA infants with a high plasma cortisol level.
[33] Doyle, L.W.; Ehrenkranz, R.A.; Halliday, H.L. Early (< 8 days) postnatal corticosteroids for preventing chronic lung disease in preterm infants. Cochrane Database Syst Rev. 2014: 5, CD001146.
Our conclusions were changes as follows.
Lines 308–311: GA is an independent determinant of plasma cortisol levels at birth. SGA infants had a high plasma cortisol level at birth; speculating that a high plasma cortisol level at birth may be associated with abnormal neurological outcome. Further follow-up studies using a large cohort are required to confirm this result.
- Under Statistical analysis Study 2, the sentence should be re-written for clarity and possibly expanded into more than one sentence.
Response: We are sorry for a vague sentence. The sentence was changed as follows.
Lines 138–141: Study 2: To compare the following between SGA and non-SGA infants, the Mann–Whitney U test and Fisher’s exact test were performed; maternal and neonatal clinical factors, cord blood plasma cortisol levels, plasma cortisol levels at 1 month of age, and Δ cortisol levels.
- In results, line 157, I believe there is a typo and you did not intend to write "i.e.".
Response: "i.e." was changed to “and”. Thank you.
- I find Table 2 and Table 3 difficult to read. There should be clearer separation of sections A., B., and C. Also, the heading for each should include "cortisol", so the reader can easily know what comparisons are being made.
Response: In Tables 2 and 3, Sections A, B, and C were clearly separated. “Cortisol level” was added in the heading.
- In the discussion, line 209 you wrote "post-growth". This should be corrected for clarity.
Response: We are sorry for our mistake. "post-growth" was deleted.

Round 2
Reviewer 2 Report
Dear authors,
thank for the revision of your manuscript. I have no further comments, but a few minor spell checking, which will be dealt with during the further editorial process.